# The Relationship between the Infrared Eye Temperature of Beef Cattle and Associated Biological Responses at High Environmental Temperatures

**DOI:** 10.3390/ani14192898

**Published:** 2024-10-08

**Authors:** Musadiq Idris, Megan Sullivan, John B. Gaughan, Clive J. C. Phillips

**Affiliations:** 1Department of Physiology, Faculty of Veterinary and Animal Sciences, The Islamia University of Bahawalpur, Bahawalpur 63100, Pakistan; 2School of Agriculture and Food Sciences, Gatton Campus, The University of Queensland, Gatton, QLD 4343, Australia; megan.sullivan12@outlook.com (M.S.); j.gaughan@uq.edu.au (J.B.G.); 3Curtin University Sustainability Policy (CUSP) Institute, Faculty of Humanities, Curtin University, Perth, WA 6845, Australia; clive.phillips@curtin.edu.au; 4Institute of Veterinary Medicine and Animal Sciences, Estonian University of Life Sciences, Kreutzwalki 1, 51014 Tartu, Estonia

**Keywords:** beef cattle, eye temperature, heat stress, heat load condition, infrared thermography

## Abstract

**Simple Summary:**

Cattle are increasingly exposed to high temperatures with accelerating climate change, presenting the risks of heat stress, poor welfare, and even death. In Australia, the sudden onset of extreme conditions means that deaths are not uncommon. Understanding the impact of high environmental temperatures on cattle behaviour and physiology is critical. We aimed to investigate changes in infrared eye temperature and their association with behavioural responses, such as changes in body postures, alterations in feeding behaviour, and body self-care. Twenty-four Angus steers were individually housed to assess heat load responses when they were fed either a standard finisher diet based on cereal grains or an alternative diet with 8% of the grains replaced by an isoenergetic amount of lucerne hay. No differences were observed between different dietary cohorts. The infrared eye temperature (IRT-Eye), of the steers exposed to high environmental temperatures was increased, and this had a strong association with behaviour and physiology. Cattle with increased IRT-Eye temperature showed stress-related behavioural responses, including a downward-facing head, ears directed backwards, and other indicators of heat stress such as increased panting, standing, and increased rumen temperature. The strong association of IRT-Eye with behaviour and physiology suggests that it can be used to detect heat stress in cattle.

**Abstract:**

Cattle in regions with high ambient temperatures are at risk of heat stress. Early detection is important to allow action to be taken to minimise the risks to cattle exposed to thermal stress. This study aimed to investigate the impact of heat stress on IRT-Eye temperature and its association with the behavioural and physiological responses of heat-stressed Angus steers (n = 24) on finisher and or substituted diets. Overall, 2 cohorts of 12 Angus steers were individually housed in a climate-controlled facility to examine responses to heat stress when fed on a standard finisher diet, based on a high percentage of cereal grains, and on a substituted diet in which 8% of the grains were replaced by an isoenergetic amount of lucerne hay. Exposing feedlot cattle to hot environmental conditions increased IRT-Eye temperature, which had a strong association with behaviour and physiology. There was no evidence of differences between the different dietary cohorts. The cattle with increased IRT-Eye temperature showed stress-related responses, including a downward-facing head, ears directed backwards, and other indicators of heat stress such as increased panting, standing, and increased rumen temperature. The strong association of IRT-Eye temperature with stress-related behaviours, as well as with rumen temperature and panting behaviour, highlights the potential for IRT-Eye to be utilised as a non-invasive tool to assess cattle responses in hot conditions.

## 1. Introduction

Hot environmental conditions are a significant welfare and production problem for livestock, especially in beef farming industries. This is especially true in tropical and subtropical regions of the world, where animals may be constantly exposed to hot conditions for prolonged periods [1,2,3]. An increased core body temperature is a major indicator of heat load [3,4,5], with cattle attempting to maintain homeostasis through physiological and behavioural responses [1]. Their core body temperatures range from 38.0 to 38.5 °C under thermoneutral conditions [2]. Body temperature is usually recorded from the rectum, rumen, vagina, or tympanic membrane of the ear [6,7,8], all differing in their accuracy and degree of invasiveness [6]. Some methods may require the handling of animals in confined places, which may result in stress to the animals and increased body temperatures [9,10]. In feedlots in Australia and the US, the death of cattle is associated with heat waves that occur during the summer period (e.g., [11]).

Infrared thermography (IRT) is widely used as a non-invasive, passive, and remote method to assess animal responses during stressful conditions. Practically, IRT has proved to be a useful non-invasive tool in veterinary and animal science for measuring fluctuations in surface temperature that may correlate with core body temperatures [1,10]. Infrared thermography has also been used as a tool for the detection of febrile conditions related to inflammation [12] and for early disease detection [13,14]. At high environmental temperatures, the animal sees increased heat loss from the skin surface through vasodilatation and increased peripheral circulation [1,15]. Infrared thermography can detect these changes [16]. The measurement of eye temperature using infrared thermography is a useful indicator of pain and stress in animals [1] during dehorning [17], bovine viral diarrhoea [13], castration [18], foot and mouth disease [19], and periods of anxiety [20]. This may be due to the position of the eye, being closer to the emotional processing centres of the brain hemispheres. Secondly, eyes have rich capillary beds, supplying blood, that are highly innervated by the sympathetic nervous system. These directly influence body surface temperature [18]. Infrared thermography can be a promising and reliable technique for assessing cattle responses during heat load conditions, with the added advantages of minimum stress to cattle and quick measurement [21,22].

Finisher or grain-based diets are usually fed to beef cattle as they are high in energy, promoting rapid weight to achieve high production levels in the beef industry. However, in heat-stressed cattle, grain-based diets may reduce rumen pH, increasing stress. In extreme cases, cattle become lame, ataxic, and uncoordinated [23,24]. Fiber-based diets are thought to improve rumen health, assisting in efficient digestion. The presence of adequate amount of fibre in diets is necessary to maintain optimal rumen pH by stimulating salivation during bolus chewing [25,26] in cattle, reducing heat load.

To define the relationship between infrared eye temperature and the animal’s responses to heat load conditions, the impact of heat load on the biological responses of feedlot cattle was studied to investigate the hypothesis that hyperthermic conditions will influence eye temperature, and that this will be associated with the behaviour and physiology of feedlot cattle. Following adverse reactions to heat stress in the first cohort, the diet was changed for the second cohort, allowing us to explore whether a substituted diet (grain substituted with forage) influenced cattle responses to heat stress, as evidenced by changes in infrared eye temperature.

## 2. Materials and Methods

### 2.1. Animal Housing

In the initial feedlot phase, animals in each cohort were kept in a feedlot pen of 162 m^2^ (27 × 6 m), with an east–west alignment. Each contained a concrete feed bunk, water troughs, and a shaded area of 1.3 m^2^/animal at midday. Animals were then randomly assigned to individual pens (each 10 × 3.4 m) in an outdoor facility containing a concrete apron, a feed bunk, water troughs, and a shaded area. In the next phase, animals were housed in two climate-controlled rooms (each 10 × 10 m), separated by a concrete wall. Each climate-controlled room was further divided into two chambers by a clear Perspex wall, with 3 pens per chamber. The ambient dry-bulb temperature, humidity cycles, and lighting schedule in each room were programmed automatically. The ventilation rate inside each CCR was maintained with a centrally controlled air-conditioning system (Table 1). Oxygen, ammonia and carbon dioxide were monitored at 0600, 1200 and 1800 h (and at midnight on HOT days) each day, using a hand-held gas meter (Gas Micro Alert, BW technologies, Honeywell, Charlotte, NC, USA). Lighting was set at 10% of maximum from 1900 to 0500 h, and to maximum at all other times. The individual pens (2.5 × 2.5 m) each had a rubber mat floor over a steel grill, which facilitated the drainage and cleaning of the pens. Pens were cleaned daily at 0630–0730 h, prior to feeding at 0900 h, by hosing all excrement from the mats and pen flooring. All animals were provided with individual access to a water trough and a feed trough (each 500 × 500 × 500 mm) inside the climate-controlled facility. The climate-controlled facility was equipped with two cameras (CW214H; K-guard Info. Co., Ltd., New Taipei City, Taiwan) above each pen that were connected to a digital video recorder (LG, XQ-L900H; T.O.M Technology Co., Ltd., Yeouido-dong, Seoul, Republic of Korea) for the surveillance of the animals.

### 2.2. Animal Management

Steers were immunised ten days before entry to the experimental feedlot facility for tick fever (trivalent chilled, 3 germ; *Anaplasma centrale*, *Babesia bovis* and *Babesia bigemina*; Department of Agriculture, Fisheries and Forestry (Biosecurity Queensland, Longreach, Australia)), clostridial diseases (5-in-1 enterotoxaemia, [pulpy kidney disease], malignant oedema, Black’s disease, tetanus, and blackleg; Pfizer Animal Health, Sydney, Australia), and for external and internal parasites (150 g/L hydrocarbon liquid; Cydectin, 5 g/L moxidectin solvent, Fort Dodge Australia P/L, Baulkham Hills, NSW, Australia). We administered bovine respiratory disease vaccination (Bovillis MH, inactivated *Mannheimia haemolytica*; Coopers Animal Health, Macquarie Park, Australia) and bovine ephemeral fever vaccination (BEFV) on d 20 of the feedlot phase and administered a second dose 24 days later before animals left the feedlot pen. Animals were inserted with a hormonal growth promoter (HGP) implant (Synovex^®^ containing trenbolene acetate, Zoetis, Parsippany, NJ, USA) on entry into the feedlot. Steers in cohort 1 were offered a starter diet of concentrates for the first 8 d in the climate-controlled facility and then an intermediate diet for 6 d. They then transitioned to a finisher diet over the next 3 d, which was used until the end of trial (Table 2). Due to the adverse heat stress response of some cattle in the first cohort, the second cohort was fed a substituted (high-fibre) diet from the second day of the HOT period and were transitioned back to the finisher diet over four days during the recovery thermoneutral period. Individually housed cattle were fed their diet at 2.5% of body weight on a DM basis, with refusals removed and weighed each morning prior to the provision of 50% of the ration at 0900 h and the remainder at 1300 h. The feed dry matter content was determined by oven drying. The animals were provided with ad libitum water during the study, and water consumption in the CCR was recorded at the time of each observation using endurance multi-jet turbine water meters (RMC Zenner, Eagle Farm, Australia).

### 2.3. Animals and Treatments

Twenty-four (24) healthy yearling Black Angus steers were procured for the study from a commercial property in Armidale, New South Wales, and transported to The University of Queensland (UQ), Gatton Campus’ Queensland Animal Science Precinct (QASP) (27.56° S, 152.34° E; 99 m elevation above sea) during the Southern Hemisphere’s summer. The steers had an initial live weight (non-fasted) of 493 ± 6.8 kg. During summer, the climate at the location is hot, humid, and sub-tropical. The maximum and minimum ambient temperatures recorded during the time of the study at the location were 30.98 ± 1.48 to 16.4 ± 2.42, respectively. A randomised design was used for the study, with the steers randomly divided into 2 cohorts of 12 animals to run separately through the climate control room, which only had capacity for this number of animals. The first cohort was fed a standard feedlot ration based on cereal grains (the finisher diet), and the second received the same diet but with 8% of the grains by mass replaced by an isoenergetic amount of lucerne hay (the substituted diet). The detailed composition of the diet is presented in Table 2.

Each cohort of 12 steers was kept for 50 d in a feedlot pen and then for 10 d in individual outdoor pens to allow acclimatisation to the handling and feeding management practices that they would receive in the climate rooms. The animals were moved to the experimental facility where cattle were kept in two climate-controlled rooms (CCR) for 18 days. The climatic conditions were changed over five different periods:Thermoneutral conditions before hot conditions (TN; d 3–4)Transition period to HOT (TP1; d 5)Hot conditions (HOT; d 6–12)Transition period to recovery (TP2; d 13)Thermoneutral conditions after hot treatment (recovery; d 14–17)

In the initial thermoneutral (TN) and recovery, the ambient dry-bulb temperature (T_A_) and relative humidity (RH) in CCR were maintained at 20 °C and 65%, respectively. The thermoneutral period transitioned to the hot climatic phase (HOT) over 1 d in TP1. In the HOT, T_A_ and RH increased each day from 0700 h to reach a maximum at 1100 h, which was maintained until 1600 h and then decreased hourly from 1600 h to the daily minimum T_A_ and RH at 2000 h. The ambient temperature declined over the HOT period before cattle entered the recovery period (Table 1). Animals were continuously monitored for real-time body temperature, panting score, respiration rate, and animal behaviour to assess the impact of a high heat load. An action plan was devised with a specific parameter set to identify animals that were not adequately coping with the conditions, as outlined below.

Ethical approval (AEC Approval No. SAFS/460/16) was obtained from The University of Queensland Animal Ethics Committee for this study, which was opportunistically added to another study of metabolic responses to heat stress. An intervention protocol was in place to mitigate severe heat stress based on animal responses to the heat stress. Monitoring was conducted every hour throughout the hot period, including the entire night. If animals exhibited signs of severe distress (constantly excessive panting, open-mouth breathing, elevated core body temperature, etc.), mitigating measures were immediately applied, including providing access to ventilation and water spray. Despite these mitigating measures, there was one sudden death on the first day of the hot period in the first cohort of animals. A second animal from this cohort was closely monitored for the whole of the first night of the hot period. Then it was removed from the study and euthanised to prevent further suffering. Following the change in ambient temperature for the second cohort, two animals were removed from the experiment due to their adverse responses to hot conditions, but their conditions were not serious and they did not have to be euthanised.

### 2.4. Behavioural and Other Key Observations

Respiratory behaviour was recorded 7 times a day, every 2 h from 0600 to 1800 h during TN and recovery and every hour over 24 h during the HOT period. A team of five trained observers recorded respiration rate and panting score (PS), with only one observer recording measurements at any one time (Table 3). The respiration rate (RR) was determined by timing the time taken for the animal to take ten breaths, as observed from flank movements. This was then converted into breaths per minute. The panting scores (PSs) of animals were visually scored using a modified scale from 0 to 4.5, where 0 indicates no heat stress and 4.5 represents a severely heat-stressed animal [27,28,29,30,31].

From the video recordings, standing, lying, the stepping of each limb, eating, ruminating, grooming, and scratching were continuously recorded using event-logging software (BORIS v. 6.0.4 [32]) for 24 h (5 min/h) on d 3 (TN), d 6–12 (HOT), and d 14–16 (recovery) (Table 3). Head, ear, and tail positions were also recorded at 5-min intervals every h for 24 h. Not all behaviours could be observed at all times: the accurate recording of ear position was not possible when the animal was ruminating, eating, drinking, scratching, or grooming; the recording of head position was not possible during eating, drinking, grooming, or scratching; and tail position could not be recorded during defecation, urination, eating, drinking, grooming, or scratching.

**Table 3 animals-14-02898-t003:** Ethogram for recorded behaviours.

Item	Description
Standing	Animal standing with limb positioned upright
Lying	Animal resting on the floor with their limb laterally or sternally recumbent
Eating	Animal consuming feed at the trough
Rumination	Animal chewing the cud or regurgitating bolus
Respiration rate	Number of breaths in a 1 min period
Panting score	Animal visually scored for the extent of panting based on a 0 to 4.5 score scale
Grooming	Animal licking any part of the body or striking one part with another part of the body
Scratching	Animal rubbing or striking any part of the body against the fixture of the pen
Ear positions	
Ear raised	Both ears being held upright above the neck with the ear pinnae facing forwards or to the side
Ear forward	Both ears’ pinnae directed forwards in front of the focal animal and held horizontally
Ear backward	Both ears being held backwards on the focal animal’s head
Ear downward	Both ears being loosely hung downwards, falling perpendicular to the head
Ear specific	Both ear pinnae (right and left) being oriented in opposite directions, or perpendicular to head rump axis, thus failing to satisfy raised, forward, backward, or downward ear positions
Head positions	
Head raised	The head held upright above withers or body top-line
Head neutral	The head held horizontally at the level of withers or body top-line
Head downwards	The head held downwards below withers or body top-line
Stepping	
Front right (FR) limb	Animal raising a front right limb and replacing it forthwith on the surface of the pen
Front left (FL) limb	Animal raising a front left limb and replacing it forthwith on the surface of the pen
Back right (BR) limb	Animal raising a back right limb and replacing it forthwith on the surface of the pen
Back left (BL) limb	Animal raising a back left limb and replacing it forthwith on the surface of the pen
Tail positions	
Tail raised	Tail held in a fixed position, held at than 45 degrees from the vertical
Tail vertical	Tail hanging downward from the vertical line of body and relaxing with no movements
Tail swishing	Swift movement of the tail in any direction around the hind quarters from its base in a side-to-side flicking manner
Tail tucked	Tail held tightly pressed in a fixed position against the rump, with the tip of the tail tucked in behind the hind limb

Adapted from Idris et al. [30,31,33].

### 2.5. Body Surface Temperature

Body surface temperature was measured by using an infrared thermometer (TN410LCE, Zytemp, HsinChu, Taiwan) on the head (midway between the eyes), shoulder (cranio-lateral surface of the thoracic cavity at the site of attachment of the forelimb), lower leg (halfway between the fetlock and hock joint), and rump (the dorso-caudal aspect of the body corresponding to the gluteal muscles) of each animal at a 1 m distance.

### 2.6. Rumen Temperature

Rumen temperature ($T_{RUM}$)-monitoring boluses (Smartstock, Pawnee, OK, USA) were inserted orally using a customised applicator on d 0 upon entry to the feedlot pens. Before administration, the rumen boluses were calibrated for 24 h by placing in a water bath at 39 °C. The boluses contained an active radio frequency identification (RFID) transmitter operating at 915–928 MHz, detectable up to 90 m. The real-time radio transmissions were communicated by a Yagi antenna to a base station and were then transformed into a database via proprietary software (TechTrol Inc., Pawnee, OK, USA). Individual internal body temperature $T_{RUM}$ recordings were transmitted to an on-site computer at 10 min intervals throughout the study. At each transmission, the previous 11 data points (i.e., the previous 110 min) were also collected in order to minimise data loss due to missed transmission. Individual measurements for the internal body temperature $T_{RUM}$ recordings of each animal were recorded at the time of the infrared thermal imaging of the cattle’s eyes.

### 2.7. Infrared Thermography (IRT)

Images of both eyes (IRT-Eye) of each animal were taken using an infrared thermal imager (FLK-Ti25 9 Hz, Fluke Corporation, Everett, WA, USA) at 0600, 1200, and 1800 h daily on d 3 (TN), d 6–12 (HOT), and d 14–16 (recovery). The images of the animals were taken whilst located in the pen inside climate control rooms at approximately 1 m distance from the eye. In order to maximise the usefulness of thermal images, the IRT-Eye images were captured at a perpendicular angle to the infrared thermal camera. Only usable infrared thermal images (which comprised 94% of all images) were selected for analysis. We used an emissivity value of 0.98; non-usable images (6%) were excluded (Figure 1). Minimum, maximum, and mean eye temperatures from both right and left eyes were determined using a zone analysis marker (Figure 2); however, only the maximum eye temperatures within the zone analysis were used for further analysis, as this represented the best image with which to determine stress in cattle [16,20]. Mean eye temperature was calculated from the infrared thermal eye temperatures of both eyes ((right eye + left eye)/2). The infrared thermography images obtained were analysed using proprietary software (Fluke Smart View Software, version 4.3, Fluke Corporation, Everett, WA, USA).

### 2.8. Climatic Data

Ambient temperature *(T_A_)* and relative humidity *(RH)* values were maintained inside the climate-controlled room using a cyclic air-conditioning system. These were recorded at 10 min intervals using two temperature and humidity data loggers per room (HOBO UX100-011, Onset, MA, USA), which were installed on the pen at 1.5 m from the ground. A temperature humidity index *(THI)* was calculated using the following equation, adapted from Thom [34]:*THI* = (0.8 × *T_A_*) + {[(*RH*/100) × (*T_A_* − 14.4)} + 46.4(1)
where *RH* = relative humidity in % and *T_A_* = ambient temperature in °C

### 2.9. Statistical Analyses

In total, 4 animals (2 from each cohort) were removed from the experiment due to their adverse response to the heat load at the start of the HOT period. Overall, there were 2 in the first cohort, with one of that died and a second that required euthanasia, and two in the second cohort that were just removed from the experiment. Therefore, in each dietary cohort, the data obtained from the 10 steers were analysed using the statistical software Minitab 17 (Minitab^®^ 17.3.1 Inc. Chicago, IL, USA) and Minitab 18 (Minitab^®^ 18.1 Inc. Chicago, IL, USA) for Windows. The transition days provided time for metabolic changes at the start of the hot conditions after the acclimatisation process and similarly at the start of the thermoneutral conditions after the hot conditions (TP1 and 2, respectively), and this was reflected in different levels of metabolic heat production due to the altered metabolic rate. Therefore, the data obtained during these transition phases to and from hot conditions were not included in our statistical analyses.

Two separate models were used to describe the data. The first made a comparison between the TN and the HOT period, accounting for diet (*D*; finisher and substituted diet), treatment periods [*P;* HOT and TN], and interactions. We used diet × treatment period as a fixed factor, and the animal identification (*ID*) was a random factor.

Secondly, linear relationships between climatic conditions and the infrared eye temperature (°C) of the feedlot cattle were determined using a mixed-effects model, with the animal identification (*ID*) acting as a random factor, and we used the following fixed factors: dietary cohort (*D*; finisher and substituted diets), treatment period [P; HOT and recovery], and the time (*t*) while the day (*d*) of the experiment was nested within the treatment. We also examined the interactions of diet × treatment period and diet × day. Data from the TN were used as covariate *(Cov*) values. The equation for the analysis was as follows:*Y* = *µ* + *D* + *P* + *t* + *d* (*P*) + *ID* + (*D* × *P*) + (*D* × *d* (*P*)) + *Cov* + *e*(2)
where *Y* is the expected value for biological response variables; *μ* is the expected mean value for response variables when input variables = zero and the factors are as described above; and e is the random error associated with experimental observations. Pairwise comparisons between treatment means were performed using Fisher’s test (*p* ≤ 0.05). When necessary, the Kolmogorov–Smirnov test was used to test for normal distributions of residuals. The second comparison, between the hot and recovery periods, was made to assess how quickly cattle return to thermoneutrality after experiencing heat stress and to evaluate the persistence of the effects of heat stress on eye temperature. This comparison provides insight into the animals’ recovery capacity and whether their physiological responses, including eye temperature, fully normalise after the heat stress period. The thermoneutral period was included as a covariate in the statistical models to account for baseline physiological conditions, allowing for the more accurate detection of changes specifically occurring due to heat stress and recovery. By comparing the hot and recovery periods, we could better understand the animals’ resilience and how quickly they recover, which is crucial for managing heat stress in livestock.

Spearman’s rank correlation coefficients (r_s_) with a two-tailed level of significance (*p* < 0.05) were determined. The data were examined using a multivariate general linear model with forward-selection stepwise regression (α to enter variable = 0.25) in order to investigate the association of infrared eye temperature with the differential means (HOT-TN) of the respiration rate, the panting score (PS), the rumen temperature, the surface temperature (ST) of the body (head, shoulder, leg, and rump), the stepping, the posture (standing, lying, head, ear and tail positions), the self-care (grooming and scratching), and the activity (eating, drinking and rumination) of animals.

## 3. Results

### 3.1. Initial Thermoneutral Period vs. Hot Period

The infrared eye temperatures of the feedlot cattle (*n* = 20), exposed first to TN and then to the high-temperature environment, are compared in Table 4. The infrared thermographic temperature (°C) of right, left, and mean infrared eye temperatures of both eyes in the feedlot cattle was greater in the HOT period than in the TN. No difference in infrared eye temperature (°C) was observed in the different dietary cohorts during the HOT treatment period. No difference was observed in the infrared R/L eye ratio (°C) of feedlot cattle during different treatment periods and diets, or interaction between the treatment period and diets.

### 3.2. Hot Period vs. Recovery Period

The infrared eye temperature in °C (right eye, left eye and mean infrared eye) of the cattle was greater in the HOT period than in the recovery period (Table 5). A significantly lower infrared eye temperature (right eye, left eye and mean infrared eye) was observed at 0600 h, increasing to a maximum value at 1200 h and then declining at 1800 h, reflecting changes in the ambient temperature (Figure 3). No significant difference was observed in the infrared eye temperature of the right eye and left eye or in the mean infrared eye temperature (°C) of beef cattle on different diets or subjected to variations in period and day. The R/L eye ratio was not affected by period, time, diet or the relevant interactions.

Using stepwise regression analysis, it was determined that the differentials in infrared eye temperature (HOT—TN) were positively correlated with HOT—TN differentials in panting score, standing, eating, vertical tail, backward and downward ear positions, and downwards and neutral head positions. They were negatively correlated with HOT—TN differentials in head and shoulder surface temperatures, scratching, stepping by the back left and front right limb, forward ear position, and raised head position.
*Infrared eye temperature* (°C) *differential* = 0.830 (±0.0025; *p* = 0.002) + 0.360 *PS* (±0.00081; *p* = 0.001) + 0.88 *HPD* (±0.0082; *p* = 0.006) + 2.22 *HPN* (±0.0070; *p* = 0.002) + 8.00 *Eating time* (±0.017; *p* = 0.001) + 2.00 *Standing* (±0.0063; *p* = 0.002) + 0.370 *EPB* (±0.0045; *p* = 0.008) + 16.00 *EPD* (±0.056; *p* = 0.002) + 7.60 *TV* (±0.0088; *p* = 0.001) − 6.00 *EPF* (±0.0028; *p* ≤ 0.001) − 0.19 *BL* (±0.00044; *p* = 0.002) − 0.11 *FR* (±0.00043; *p* = 0.002) − 14.70 *Scratching* (±0.027; *p* = 0.001) − 0.081 *Shoulder temperature* (±0.00017; *p* = 0.001) − 0.010 *Head temperature* (±0.00018; *p* = 0.01) − 854 *HPR* (±0.75; *p* = 0.001)(3)
where *PS* = panting score; *HPD* = head downwards; *HPN* = head neutral; *EPB* = backward ear; *EPD* = ear position downward; *TV* = tail vertical; *EPF* = forward ear, *BL* = back left stepping; *FR* = front right stepping; and *HPR* = raised head.

The quadratic relationship of rumen temperature with IRT-Eye temperature (°C) is expressed by the equation below and presented in Figure 4:*Rumen temperature* (°C) = 88.3 − 3.2 *IRT-Eye temperature* (°C) + 0.05 *IRT-Eye temperature*^2^ (°C) (*r*^2^*_adj_* = 70.7%; *F_value_* = 672; *p* ≤ 0.001)(4)

To determine if there were any other non-linear relationships, several non-linear terms were tested, but they were not significant and/or the r^2^ values were not increased.

Spearman’s rank correlations of the differences in the HOT—TN means of infrared eye temperature (°C) with differences in the HOT—TN means of behaviours were used to determine the behavioural impact when IRT cattle responded the most. Change in IRT-Eye temperature was positively associated with change in head downward position (correlation coefficient (CC) 0.481; *p* = 0.03), ear backward (CC 0.54; *p* = 0.01), tail vertical (CC 0.68; *p* ≤ 0.001), standing (CC 0.51; *p* = 0.02), front right limb stepping (CC 0.503; *p* = 0.05), panting (CC 0.51; *p* = 0.02); and rumen temperature (CC 0.56; *p* = 0.01). It was negatively correlated with scratching (CC—0.60; *p* = 0.01) and grooming (CC—0.44; *p* = 0.05).

## 4. Discussion

The results for the behavioural responses of beef cattle to hot conditions are detailed in the work of Idris et al. [31]. In this manuscript, the impact of heat load on biological responses of feedlot cattle was studied to investigate the hypothesis that hyperthermic conditions influence eye temperature, which is thought to be associated with the behaviour and physiology of feedlot cattle. We further explored whether a substituted diet (grain substituted with forage) would influence cattle responses to heat stress through any changes in the infrared eye temperature. Behavioural responses that were related to the animals’ ability to cope with the hot conditions were associated with increased IRT-Eye temperature (°C), reflecting its importance for possible applications in feedlot industry.

Hot climatic conditions can serve as a potential stressor to beef cattle, sometimes leading to low production and high mortality rates in feedlots. Cattle are susceptible to hyperthermia during hot environmental conditions and when their thermoregulatory responses of animals are insufficient to maintain thermoneutrality, resulting in a rise in core body temperature [35], a major indicator of hyperthermic conditions. Earlier efforts to use IRT to assess heat stress in feedlot cattle through the association of the infrared body surface temperature with the rumen temperature failed to predict feedlot cattle responses to hot conditions [10]. However, the association of infrared eye temperature (°C) with rumen temperature in the current study demonstrates the value of IRT-Eye measurements as an alternative method for measuring core body temperature (Figure 4). Consequently, infrared thermography may be a promising and reliable technique for assessing cattle responses and well-being [21,22] during hot days, with the additional advantage of minimum stress and quick measurement.

The rise in core body temperature during heat stress provokes animals to increase their peripheral circulation [15] to dissipate accumulated heat from the body. This change in the blood circulation leads to increased heat dissipation from the body’s surfaces, including the eyes. This is especially the case around the *caruncula lacrimalis,* which is innervated by rich capillary beds [16]. Not all surface temperatures are correlated with body temperature (e.g., limb surface temperature); however, infrared eye temperature has been shown to have a correlation with core body temperature [35,36]. Therefore, infrared thermography can determine changes in peripheral circulation through changes in heat loss mechanisms as a non-invasive, passive, remote, and quick tool for assessing animal responses.

In our cattle, increases in the infrared thermographic temperature (°C) of the right eye and left eye and in the mean eye temperature of both eyes were observed in the HOT period compared with the TN period and recovery periods (Table 4 and Table 5). Similarly, IRT-Eye was at its maximum at 1200 h when the dry-bulb ambient temperature was at its daily maximum limit during hot days and dropped following a decline in ambient temperature (Table 4; Figure 3). The rise in body temperatures (rectal and rumen temperature) in cattle exposed to hot environmental conditions is well established (e.g., Gaughan [15]; Brown–Brandl et al. [37]; Lees [38]). Stewart et al. [39] assessed IRT-Eye temperature in dairy cattle as a measure of pain and stress during handling. Further, Church et al. [40] studied the impact of environmental factors on the IRT-Eye temperature of cattle. In other studies, the infrared thermography of eyes has widely been reported as a tool for the detection of thermal conditions related to inflammation [13], dehorning [19], castration [20] and disease detection [13,14]. Recently, Shu et al. [41] reported the eye to be the best region for measuring thermal variation in order to assess the heat stress response in dairy cattle. Similarly, in our study, the rise in the IRT-Eye temperature in the cattle during the HOT period confirms the potential of the IRT-Eye to be used as a useful tool (in addition to rectal and rumen temperature) for assessing thermal variations in cattle experiencing hot environmental temperatures.

No difference in infrared eye temperature (in °C) was observed in different dietary cohorts during the HOT treatment period. The absence of significant differences in the IRT response between the two dietary cohorts could be for several reasons. One possibility is that, although the diets differed in composition and were nutritionally important, the variation might not have been large enough to impact the animals’ ability to regulate their body heat. Both diets provided energy and or fibre levels that provided for the animals’ requirements. Secondly, infrared thermography could only measure the body surface temperature [1] and it was unable to capture the internal adaptations inside the body, especially minor perturbations. In summary, we suspect that the dietary differences between cohorts were not enough to result in physiological changes that would result in differences in body surface temperature that could be measured through IRT.

In the current study, a negative association of IRT-Eye temperature differentials existed with temperature differentials in head and shoulder surfaces, demonstrating less difference in body surface temperatures with increasing IRT-Eye temperature. Thus, cattle that had the biggest IRT-Eye temperature response saw the smallest change in peripheral circulation in hot conditions. This probably reflects natural variation in the ability of cattle to respond to heat through evaporative cooling from the skin’s surface, enabling them to dissipate accumulated heat during hot conditions [42,43]. Those with a limited ability to do this had the greatest increase in core body temperature, as evidenced by increased IRT-Eye values. No evidence of infrared eye temperature or lateralised IRT-Eye responses was obtained, despite the fact that cattle show lateralised visual preferences during various stressful conditions [44,45]. However, using an eye more does not mean that its temperature is expected to increase. In the current study, cattle exhibited compensatory behaviours (e.g., increased standing time, reduced activity such as grooming, scratching) that might have minimised the impact of diet on thermoregulation, resulting in similar eye temperatures across both groups.

During hot environmental conditions, cattle attempted to cope with hot conditions by reducing the body heat load through various behavioural and physiological adjustments in order to maintain a constant body temperature [46]. Adjustments made to maintain body temperature apparently lag three to four hours behind changes in daily dry-bulb ambient temperature [15,47]. However, the exact delay can vary; in some cases, respiratory signs might appear sooner if the animal is already in a compromised state, or later if the animal has better coping mechanisms. This time lag in apparent core body temperature may delay signs of hyperthermia until it is too late to provide support to animals [47]. Monitoring coping behaviours associated with immediate increases in body temperature can be helpful in providing necessary support to animals suffering from hot conditions. Cattle with greater increases in the IRT-Eye (°C) had smaller changes in scratching and stepping behaviour (front right and back left stepping). A decline in scratching behaviour [48] and eating [49] has been reported in dairy cows in response to increased body temperatures. Animals with higher differential of IRT-Eye measures had a greater decline in eating time, as reported earlier in our detailed behaviour studies, published as Idris [30] and Idris et al. [31]. A reduced eating time reflects a decline in feed intake to reduce the heat of digestion. This could be due to a decline in body energy that might have affected the animals’ chewing ability [49,50,51]. Increased back left and front right limb stepping have been reported to be indications of stress-related responses in sheep during simulated sea motion [52]. However, the increase in back left and front right limb stepping with increasing IRT-Eye could be an effort to attempt to escape from the stressful condition [53,54]. Ultimately, the decline in these energy-dependent activities such as scratching can be attributed to a decline in body energy resources [55] and/or to a reduction in heat production [56] in response to hot environmental conditions.

The regression analysis shows that, during the HOT period, the cattle with the biggest increase in IRT-Eye temperature were more likely to spend more time with a vertical tail, hanging stationary. A vertical tail is the most observed tail position during routine daily activities, such as walking, standing and lying, and is sometimes taken as an indication of calm and relaxed behaviour in cattle [54,57]. However, in the hot conditions of this study, the involvement of cattle in protective behaviours such as tail swishing (the opposite of vertical stationary tail), scratching, and stepping potentially helped them to cope with the conditions by creating air movement (convection) over exposed surfaces. Convection depends on the air flow and contributes about 15% to heat loss compared to conduction, which accounts for only 5% of heat loss [58]. In this study, tail swishing allows for more significant heat loss by convection associated with the vulvar region, which tends to be under the effect of extensive peripheral vasodilation. Thus, cattle not engaging in these protective behaviours had the highest IRT-Eye increases.

Among various signs of heat stress, increased body temperature (rectal and rumen), panting, and standing time have all been reported as early signs of heat stress, as cattle attempt to maintain constant body temperature by dissipating body heat [59,60]. Lees et al. [9] reported that during hot environmental conditions, cattle accumulate heat from the surrounding environment, increasing the rumen temperature which must be dissipated from the body. The animal increases heat loss from the skin surface through vasodilatation and increased peripheral circulation [1,15]. This includes the eyes, especially around the *caruncula lacrimalis,* which is innervated by rich capillary beds [16]. Eyes are not usually influenced by ambient temperature due to the rich blood supply and high level of innervation by the sympathetic nervous system, which directly influence body surface temperature. There is a strong correlation between infrared eye surface temperature and core body temperature [16,18]. In the current study, the quadratic relationship between rumen temperature and eye temperature reinforces earlier findings of a strong correlation of IRT-Eye with core body temperature (i.e., rumen temperature in our study). Furthermore, rumen temperature increases rapidly at high core body temperatures, probably due to the high environmental temperatures. Regarding the relationship between eye temperature and rumen temperature, it is important to acknowledge the limitations of rumen temperature as a proxy for core temperature. Rumen temperature can be influenced by factors like feed intake and water consumption, which may cause discrepancies when compared to rectal or vaginal temperature measurements [16,18]. IRT-Eye measurements have an advantage, being non-invasive and minimising stress; however, they may not always outperform easily observable indicators like respiration rate (RR) or panting score (PS) in field settings where equipment may not be available. The importance of infrared thermography cannot be neglected in the industry for the continuous monitoring of large herds via the installation of automated thermal cameras in designated areas.

Standing helps by increasing the surface area of the body for maximum evaporation from the skin surface [33], particularly the naked skin underneath the torso, and similarly panting increases heat dissipation through respiratory channels and the evaporation of saliva from the tongue surface, which is highly innervated with blood vessels [37]. In the current study, the association of the infrared eye temperature of cattle with rumen temperature (Figure 4), standing, and panting during hot conditions further supports the inclusion of IRT-Eye temperature in the list of the tools that can be used as major indicators of heat stress in beef cattle.

The cattle with the most increased infrared eye temperatures in the current study showed a greater increase in the time spent with their heads downward, more ears backward, showed a greater decline in neutral head position, and had ears downwards during hot conditions. There was less change in the time spent with a raised head and forward ear position. Increased head downward [54,59] and backward ears [61,62] have been previously reported in cattle and sheep during various stressful activities, as compared with normal behaviours, which result in a neutrally positioned head and forward, downward, or axial ear positions. The behavioural changes in the orientation of the ears, head, and tail have already been reported in Idris et al. [31]. This downward head position (below the withers) may indicate discomfort [52,59], while holding the head above or at the level of the withers indicates healthy, active, and vigilant animals [63]. Cattle spend an increased amount of time with their head in a downwards position following castration [64] and when handling stress [59]. Cattle respond to heat with more time spent with their head downward [30,31]. The greater downward ear position response in the current study could be attributed to low energy reserves in animals due to heat stress. There are inconsistent reports regarding ear positions, as backward and forward ear position have also been reported during feeding and grooming (brushing) activities [55]. Similarly, forward ears have also been reported as the predominant ear position, suggesting reactive/anxious behaviour [65]. These inconsistent results for ear positions can probably be attributed to different arousal levels in animals and their association with both positive and negative valence [61].

The association of IRT-Eye temperature with previously reported stress-related responses, such as downward head [52,59], backward ears [60,61], and other indicators of heat stress—increased panting [15], standing, and rumen temperature [9]—rather than normal relaxed cattle behaviours (such as scratching), shows the importance of IRT-Eye measurements as non-invasive tools to assess cattle responses to hyperthermia. The current study was performed in controlled environmental conditions where the major confounding factors reported for the use of infrared thermography, including camera settings (object localisation, distance, humidity, and emissivity), solar radiations, and air velocity, were closely controlled. However, in the field conditions, these can be overcome by the installation of automated thermal cameras in areas with no direct exposure of solar radiations and minimum air velocity [16]. The automated thermal imagers with an ability to localise the eye [66] would best for the measurement of the infrared eye temperature of cattle in field conditions. Future studies should explore the effects of dietary interventions on a larger and more diverse sample of cattle across varying climatic conditions. Moreover, investigating other physiological and behavioural responses to heat stress would help validate the efficacy of different dietary strategies to mitigate heat stress. Additionally, while eye surface temperature was used as an indicator of thermal status, other regions of interests for infrared thermography, such as the animal’s flank or using body surface temperature at multiple sites, could provide a more comprehensive assessment. Future studies should explore the effects of dietary interventions on a larger and more diverse sample of cattle across varying climatic conditions, but with careful control of the onset of hot conditions. Moreover, investigating other physiological and behavioural responses to heat stress would help validate the efficacy of different dietary strategies to mitigate heat stress.

## 5. Conclusions

Cattle suffering from heat stress had higher mean infrared eye temperatures of both eyes, with no difference in response between those fed finisher and substituted diets. Infrared eye temperature (°C) was strongly associated with already validated stress-related behaviours and reported indicators of heat stress in cattle. There was no evidence of lateralised eye temperature responses in cattle. Based on changes in infrared eye temperature (°C) during hot conditions and their association with the rumen temperature and panting behaviour, as well as with other coping behaviours, infrared eye temperature could be combined with these related behaviours and utilised as a non-invasive tool in the feedlot industry during hot environmental conditions. The strong association of IRT-Eye with behavioural and physiological responses suggests that IRT-Eye can be used as an indicator to recognise heat stress in beef cattle.

## Figures and Tables

**Figure 1 animals-14-02898-f001:**
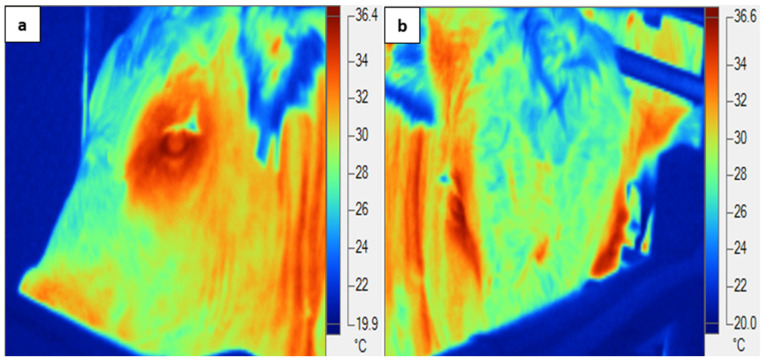
Examples of (**a**) useable and (**b**) unusable infrared thermal images of the eye.

**Figure 2 animals-14-02898-f002:**
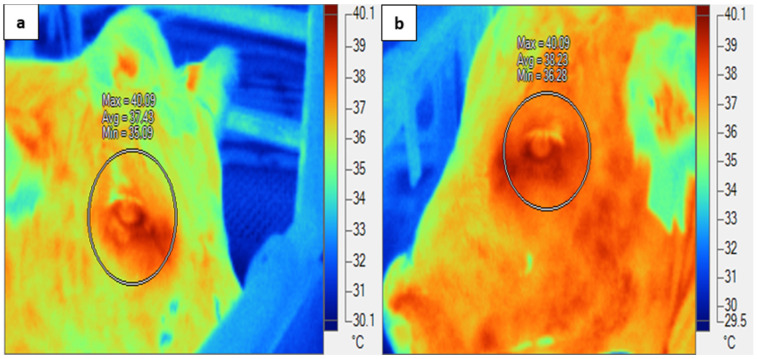
Zone analysis of the eye: (**a**) right eye; (**b**) left eye.

**Figure 3 animals-14-02898-f003:**
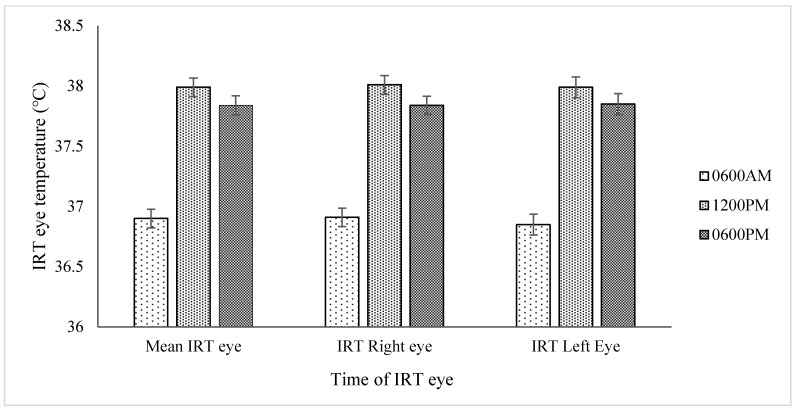
Infrared thermographic eye temperature (°C) of both eyes, the right eye, and the left eye at different times of the day (*p* ≤ 0.001).

**Figure 4 animals-14-02898-f004:**
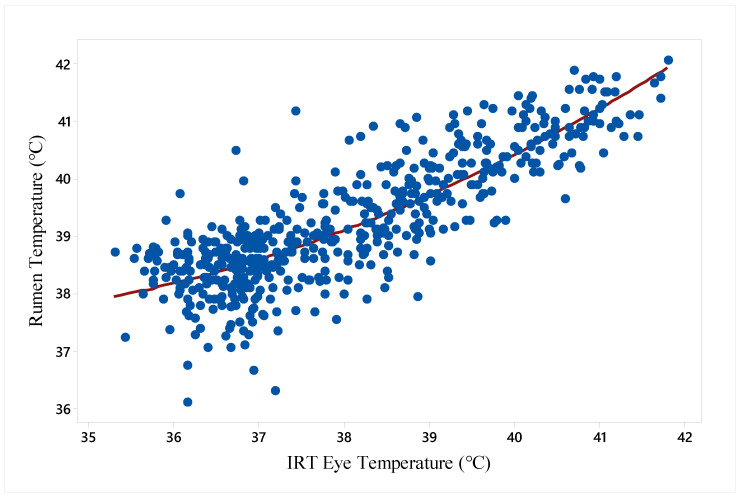
Quadratic relationship between rumen temperature and infrared thermographic eye temperature (°C). {Rumen temperature (°C) = 88.3 − 3.2 IRT-Eye temperature (°C) + 0.05 IRT-Eye temperature^2^ (°C) (r^2^_adj_ = 70.7%; F_value_ = 672; *p* ≤ 0.001)}.

**Table 1 animals-14-02898-t001:** The ambient temperature, relative humidity, and temperature humidity index for cattle (*n* = 12) when they were in the climate control facility.

Day	Treatment Phase	Min T_A_ (°C)	Max T_A_ (°C)	Mean T_A_ (°C)	Min RH (%)	Max RH (%)	Mean RH (%)	Min THI	Max THI	Mean THI
0	ACC	19.7	21.0	20.1	60.9	90.6	66.2	65.5	69.1	66.3
1	ACC	19.7	21.0	20.1	60.9	90.6	66.2	65.5	69.1	66.3
2	ACC	19.5	21.6	20.0	60.0	89.3	67.1	65.3	68.9	66.2
3	TN	19.5	20.8	19.9	61.3	90.1	67.9	65.3	68.8	66.1
4	TN	19.6	24.0	20.2	60.0	89.1	68.2	65.5	72.0	66.4
5	TP1	19.9	40.5	33.2	42.9	88.4	66.1	66.3	92.6	84.9
Finisher dietary cohort—transition to 30 °C from 00.00 h on day 5 Substituted dietary cohort—transition to 30 °C from 21.00 h on day 5										
6	HOT	28.4	40.2	33.0	43.3	82.8	65.8	80.5	91.8	84.8
7	HOT	28.4	38.1	32.1	42.3	84.2	63.7	78.3	89.1	83.0
8	HOT	24.9	34.3	28.7	44.3	82.0	65.9	73.6	85.0	78.4
9	HOT	22.6	34.4	28.0	45.81	79.5	66.2	69.9	85.7	77.5
10	HOT	20.6	30.3	24.3	54.4	80.5	66.7	67.2	80.0	72.3
11	HOT	20.4	30.4	24.2	45.3	80.6	65.8	67.1	79.2	72.0
12	HOT	19.7	21.3	20.3	50.0	90.5	64.6	65.8	68.8	66.4
13	TP2	19.7	20.7	20.1	56.4	91.3	65.5	65.6	68.6	66.2
14	Recovery	19.7	21.4	20.1	58.1	89.0	66.7	65.6	69.6	66.2
15	Recovery	19.6	20.5	19.9	58.4	90.3	66.4	65.6	68.2	66.0
16	Recovery	19.4	25.0	20.5	57.8	93.5	66.4	65.2	73.2	66.8
17	Recovery	19.3	23.7	21.1	58.1	69.0	61.9	64.9	71.1	67.5

T_A_: ambient temperature (°C); RH: relative humidity; THI: temperature humidity index; ACC: acclimatisation to climate-controlled facility; TN: thermoneutral conditions before high-temperature treatment; TP1 and 2: transition phases to and from hot conditions; HOT: high-temperature treatment; recovery: thermoneutral conditions after high-temperature treatment as a recovery period.

**Table 2 animals-14-02898-t002:** Diet ingredients and nutrient composition for starter, intermediate, finisher and substituted diets fed to cattle.

Item	Starter	Intermediate	Finisher	Substituted
**Ingredients, % of diet**				
Grain-based mix *	62.1	74.5	86.8	78.7
Whole cottonseed	9.0	16.5	9.0	9.0
Lucerne hay	28.9	9.0	4.2	12.3
**Nutrient composition**				
DM, g/kg fresh weight	880	893	887	886
ADF, g/kg DM	263	257	119	177
NDF, g/kg DM	404	375	229	253
NE_g_, MJ/kg DM	29	29	30	30
ME, MJ/kg DM	116	119	132	131
DE, MJ/kg DM	143	147	163	162
Crude fibre, g/kg DM	218	197	87	124
Nitrogen-free extract, g/kg DM	503	548	678	685
Fat, g/kg DM	46	43	46	43
Feed digestibility, g/kg DM	768	791	861	868
Digestible DM, g/kg DM	676	707	763	769
Digestible protein g/kg DM	133	125	130	131
Starch, g/kg DM	229	218	432	432

* grain-based mix: steam-rolled barley 89.2%, feedlot pellet 9.2%, and vegetable oil 1.6%; feedlot pellet: wheat brain, aleurone, germ, and pollard (Millrun Wheat, Riverina, Brisbane, Australia) 55.9%, ammonium sulphate 2.6%, dry-rolled wheat 12.5%, calcium carbonate (limestone) 15.6%, Rumensin 100 0.3%, magnesium oxide 0.7%, zinc supplement (Availa zinc 100) 0.34%, vegetable oil 3.1%, salt, (NaCl) 2.8%, urea 5.7%, vitamin A 500 0.009%, vitamin E 0.057%, and mineral supplement (XFE-Select L) 0.385%.

**Table 4 animals-14-02898-t004:** Infrared eye temperature (IRT-Eye) of cattle on a finisher or substituted diet exposed to high temperatures (HOT) or an initial thermoneutral period (TN).

Parameters	Finisher Diet		Substituted Diet		SED	F-Value (1, 18; d.f. ^†^)	*p*-Value		
	TN	HOT	TN	HOT			Period (P)	Diet (D)	D × P
Mean eye temperature (°C)	36.75 ^b^	38.73 ^a^	36.91 ^b^	38.52 ^a^	0.123	429	≤0.001	0.800	0.050
Right eye temperature (°C)	36.73 ^b^	38.72 ^a^	36.91 ^b^	38.50 ^a^	0.114	492	≤0.001	0.900	0.020
Left eye temperature (°C)	36.76	38.74	36.93	38.51	0.149	287	≤0.001	0.800	0.070
Right/left eye ratio (°C)	0.999	0.999	0.999	1.00	0.0016	0.11	0.800	0.800	0.970

IRT: infrared thermography; SED: standard error of the difference between two means; HOT: high-temperature treatment period; day 6–12; TN: initial thermoneutral period before high-temperature treatment; day 3; ^†^ treatment, error degrees of freedom; D: diet; P: period. Superscripts ^a,b^; Means within each variable with different superscripts are different at *p* ≤ 0.05 by Fisher’s test.

**Table 5 animals-14-02898-t005:** Infrared eye temperature (IRT-Eye) of feedlot cattle on finisher or substituted diets exposed to high temperatures (HOT treatment) or to a recovery period.

Parameters	Finisher Diet		Substituted Diet		SED	F-Value (d.f. ^†^)	*p*-Value				
	HOT	Recovery	HOT	Recovery			Period (P)	Diet (D)	Time (t)	D × P	D × d
Mean Eye Temperature (°C)	38.73	36.58	38.52	36.49	0.241	1758 (1, 540)	≤0.001	0.300	≤0.001	0.200	0.900
Right Eye Temperature (°C)	38.73	36.59	38.50	36.52	0.247	1545 (1, 511)	≤0.001	0.300	≤0.001	0.100	0.700
Left Eye Temperature (°C)	38.72	36.57	38.52	36.43	0.242	1486 (1, 484)	≤0.001	0.300	≤0.001	0.600	0.900
Right/Left Eye ratio (°C)	1.00	1.00	1.00	1.00	0.0023	2.09 (1, 459)	0.149	0.782	0.366	0.993	0.580

IRT: infrared thermography; SED: standard error of the difference between two means; HOT: high-temperature treatment period; day 6–12; recovery: thermoneutral period after high-temperature treatment; d 14–16); ^†^ treatment, error degrees of freedom; D: diet; d: day; P: period.

## Data Availability

The data presented in this study are available on request from the corresponding author.

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
