# Peer review of "The Relationship between the Infrared Eye Temperature of Beef Cattle and Associated Biological Responses at High Environmental Temperatures"

_animals, 2024, doi:10.3390/ani14192898_

Round 1
Reviewer 1 Report
Comments and Suggestions for Authors
#60—It would be interesting to add background information identifying the reasons why such different conditions can be monitored by IV thermography (possibly mentioning the sympathetic nervous system).
#88 - in addition to the isoenergy of the diets, the authors should identify whether they also had the same % of protein. The authors should also explain the reasons for incorporating the two iso-energetic diets. It is assumed that the diets differ in calorific value, i.e. they may produce more heat during rumen digestion, but this should be explained.
#101: Considering the transition days, the authors should mention that there was time for metabolic changes due to the start of the acclimatisation process, and this could be reflected in lower metabolic heat production due to the reduction in metabolic rate.
#107: Table 1 should appear immediately after its reference in the text.
#106 - the authors should consider placing chapter 2.2 before chapter 2.1. and chapter 2.2. should become 2.3, while Chapter 2.3 would become 2.2.
#237 - the situation reported is serious, given the death of the animal. The ethics committee should have demanded contingency criteria for animals leaving the trial. Animals have different heat tolerances, and therefore, given the number of heat stress indicators assessed, the animals should have left the experiment earlier to avoid the highly penalising situation for the animals in question. Therefore, the authors should mention whether there were any exclusion criteria and why one animal died and another was euthanised. If the journal editor considers this, it can be a reason to reject the article.
#244—Given the type and diversity of the data, There is no reference to compliance with normality and homoscedasticity. The authors should add this information.
#304: Taking equation 3 into account, the authors should explain in the discussion chapter the reason for including two antagonistic head positions in the model, both of which play a large part in the final variation of the EIT.
#309: Given equation 4, which reports the relationship between rumen temperature and ocular IV temperature, the authors must discuss better the reasons for this occurrence, bearing in mind that there were no significant differences between the diets in the ocular IV temperature response.
#329—The objectives of introducing two diets into the study and identifying the alternative hypothesis need to be better explained, bearing in mind that the null hypothesis was fulfilled.
#343—The authors should include in the text how and under what conditions IV temperature measurements can be used in a feedlot context (e.g. when the animals are going to be weighed and are confined to a small space).
#258 - the authors have ruminant temperatures over circadian cycles, so it would be interesting to put these variations in graphs for the various periods.
#368—The authors should consider that there tends to be a high sweating rate in the spots mentioned, which has a probable impact on reducing the temperature of the epidermis. As the animals are not subject to solar radiation, the combined effect of evaporation and convection may likely have led to a reduction in temperature, which IR detected.
#404 - the authors should consider that the vertical positioning of the tail allows for more significant heat loss by convection associated with the vulvar region, which tends to be under the effect of extensive peripheral vasodilation.
#453—These considerations should be further developed, taking into account the practical applicability of the study's results.
The authors should consider some aspects: Wouldn't take the animals to a place where eye temperatures can be measured promote stress and sympathetic nervous system responses and thus determine an increase in eye temperature?
Given the relationships between eye temperatures and the panting score, wouldn't it be more interesting and accessible to establish a more accurate correspondence between these two variables so that the panting score can be used more accurately and efficiently?
Author Response
Reviewer 1:
Comments and Suggestions for Authors
Comment 1: #60—It would be interesting to add background information identifying the reasons why such different conditions can be monitored by IV thermography (possibly mentioning the sympathetic nervous system).
Response 1: Thank you for your insightful suggestion. We, the authors, have incorporated background information into the manuscript to explain why different conditions can be effectively monitored using infrared (IR) thermography. Specifically, we have highlighted the role of the sympathetic nervous system in regulating peripheral blood flow, which directly influences skin temperature. These temperature changes can be detected through IR thermography, making it a powerful tool for monitoring physiological and pathological conditions.
This addition aims to provide a more comprehensive understanding of the mechanisms behind thermographic monitoring and enhance the overall context of the study.
Comment 2: #88 - in addition to the isoenergy of the diets, the authors should identify whether they also had the same % of protein. The authors should also explain the reasons for incorporating the two iso-energetic diets. It is assumed that the diets differ in calorific value, i.e. they may produce more heat during rumen digestion, but this should be explained.
Response 2: Thank you for your valuable comments. The diet was formulated by a professional nutritionist considering the balance of all ingredients, in terms of their % inclusion in the diet as well as final nutrient composition of the diet. There was no difference in the percentage inclusion of protein in both diets, as presented in Table 2.
Comment 3: #101: Considering the transition days, the authors should mention that there was time for metabolic changes due to the start of the acclimatisation process, and this could be reflected in lower metabolic heat production due to the reduction in metabolic rate.
Response 3: This is a very valid point from the reviewer. Considering the transition days, we believe that there was time for metabolic changes due to the start of the hot conditions after acclimatisation process during transition period 1 (TP1) and similarly due to the start of the thermoneutral conditions after hot conditions (Recovery Period), and this could be reflected in different metabolic heat production due to the altered metabolic rate. Therefore, the data obtained during these transition phases to and from hot conditions were not included for statistical analyses. This information has been added in the statistical analyses to provide a more comprehensive understanding of the data analyses.
Comment 4: #107: Table 1 should appear immediately after its reference in the text.
Response 4: Thank you for your observation. We have adjusted the placement of Table 1 in the manuscript so that it now appears immediately after its first reference in the text. This change enhances the readability and flow of the manuscript, ensuring that readers can easily follow the discussion with the relevant data at hand.
Comment 5: #106 - the authors should consider placing chapter 2.2 before chapter 2.1. and chapter 2.2. should become 2.3, while Chapter 2.3 would become 2.2.
Response 5: Amended as suggested.
Comment 6: #237 - the situation reported is serious, given the death of the animal. The ethics committee should have demanded contingency criteria for animals leaving the trial. Animals have different heat tolerances, and therefore, given the number of heat stress indicators assessed, the animals should have left the experiment earlier to avoid the highly penalising situation for the animals in question. Therefore, the authors should mention whether there were any exclusion criteria and why one animal died and another was euthanised. If the journal editor considers this, it can be a reason to reject the article.
Response 6: Thank you for your thoughtful and detailed comments. We acknowledge the seriousness of the situation, especially given the unfortunate death of one animal and the euthanasia of another. However, we considered it important to be totally transparent in this difficulty that we faced for the study. In addition, it is important for us that this study was not the primary reason for conducting the work, which was otherwise focused on parameters not relevant to our study. Our study was an additional component that was opportunistically added on.
In response, we have revised the manuscript to include a clear explanation of the contingency criteria that were in place for the animals involved in the trial. Animals were continuously monitored for real-time body temperature, panting score, respiration rate and animal behaviour to assess the impact of high heat load. An action plan was devised with specific parameters set to identify animals that were not adequately coping with the conditions. However, the one death occurred at night and was unexpected. It could not have been predicted from the action plan. We understand the importance of animal welfare in research and regret the outcomes in these cases. We believe that the additional details provided in the revised manuscript address the concerns raised and help to clarify the protocols we employed to minimize harm to the animals.
Please let us know if further revisions or clarifications are necessary.
Comment 7: #244—Given the type and diversity of the data, There is no reference to compliance with normality and homoscedasticity. The authors should add this information.
Response 7: This is now included.
Comment 8: #304: Taking equation 3 into account, the authors should explain in the discussion chapter the reason for including two antagonistic head positions in the model, both of which play a large part in the final variation of the EIT.
Response 8: The both head position are antagonist, however, they are differential means and their reason is explained in the discussion in lines 500-505 as ‘’ The cattle with the most increased infrared eye temperatures in the current study showed a greater increase in the time spent with head downward, more ears backward and a greater decline in neutral head position and ears downwards during hot conditions. There was less change in the time spent with a raised head and forward ear position. Increased head downward’’
Comment 9: #309: Given equation 4, which reports the relationship between rumen temperature and ocular IV temperature, the authors must discuss better the reasons for this occurrence, bearing in mind that there were no significant differences between the diets in the ocular IV temperature response. #329—The objectives of introducing two diets into the study and identifying the alternative hypothesis need to be better explained, bearing in mind that the null hypothesis was fulfilled.
Response 9: We are grateful to you for highlighting these important points. The relevant section has been amended to include this extra discussion.
Comment 10: #343—The authors should include in the text how and under what conditions IV temperature measurements can be used in a feedlot context (e.g. when the animals are going to be weighed and are confined to a small space).
Response 10: The relevant information is presented at the end of discussion section as ‘’……… in the field conditions these can be overcome by installation of automated thermal cameras in areas with no direct exposure of solar radiations and minimum air velocity [12]. …………… would best for the measurement of infrared eye temperature of cattle under field conditions.’’
Comment 11: #258 - the authors have ruminant temperatures over circadian cycles, so it would be interesting to put these variations in graphs for the various periods. #368—The authors should consider that there tends to be a high sweating rate in the spots mentioned, which has a probable impact on reducing the temperature of the epidermis. As the animals are not subject to solar radiation, the combined effect of evaporation and convection may likely have led to a reduction in temperature, which IR detected.
Response 11: The changes in the pattern of rumen temperature over circadian cycles was one of the objectives of another research team member and this will be published as a separate study. Therefore, just the correlation of IRT eye with rumen is presented in the current manuscript.
Comment 12: #404 - the authors should consider that the vertical positioning of the tail allows for more significant heat loss by convection associated with the vulvar region, which tends to be under the effect of extensive peripheral vasodilation.
Response 12: Thank you for this suggestion; the relevant information has been added as suggested.
Comment 13: #453—These considerations should be further developed, taking into account the practical applicability of the study's results. The authors should consider some aspects: Wouldn't take the animals to a place where eye temperatures can be measured promote stress and sympathetic nervous system responses and thus determine an increase in eye temperature? Given the relationships between eye temperatures and the panting score, wouldn't it be more interesting and accessible to establish a more accurate correspondence between these two variables so that the panting score can be used more accurately and efficiently?
Response 13: These are indeed valuable suggestions. In the current study, using stepwise regression analysis, it was determined that the differentials in infrared eye temperature (HOT – TN) were positively correlated with HOT – TN differentials in panting score, as well as with other parameters under study. This is the best correspondence between these two variables so that the panting score can be used more accurately and efficiently in the presence of other contributing factors as presented in equation 3 of the manuscript. In future, we will definitely consider a detailed study highlighting the relationships between eye temperatures and the panting score. Other relevant suggestions have also been incorporated in relevant sections as suggested.

Reviewer 2 Report
Comments and Suggestions for Authors
Non-invasive and relatively easy methods to accurately evaluate animal health and welfare are an alternative to be used in farms. While most studies focus on behavioral assessment, increases in infrared eye thermography during heat stress have already been described in several species including domestic animals and wild animals. The present manuscript correlates this increase with other behavioral and physiological responses, including rumen and surface temperature. One of the aspects that need to be improved is the justification of evaluating two types of diets and a proper discussion on why they didn’t show significant differences in the IRT response. I left some comments below.
Abstract.
Lines 29-30. Consider improving the introductory sentence, something similar to the simple summary.
Lines 30-31. Clearly state that the aim of the study was… Additionally, I recommend adding a couple of sentences explaining the variables that the authors evaluated (e.g., which physiological parameters? Rectal temperature, heart rate, respiratory rate, etc. The same applies to the behaviors)
Lines 31-34. Somewhere in the introductory sentences of both the simple summary and the abstract, the authors need to briefly explain the relevance of using two types of diets when evaluating the responses of cattle to heat stress. For example, if diet changes have been proposed as a management strategy to improve the condition of cattle in hot climates.
Lines 34-35. When mentioning that the animals were exposed to hot environmental conditions, the authors must include that the animals were inside climate control rooms with different temperatures, as explained in lines 96-100.
Line 35. When mentioning a “strong association”, I recommend adding the values.
Lines 36-37. Here, visual lateralization is mentioned. However, there is no mention of this in the previous text. You can see my recommendation about explaining the evaluated variables before presenting the results.
Introduction
Lines 46-52. Before this paragraph, I suggest adding some information regarding the effect that heat stress has on animal health and welfare so the reader can understand the importance of evaluating the thermal state of the animals.
Lines 54-56. I consider it important to specify that, according to the species, the thermal response of certain regions or thermal windows has shown correlations with body core temperature. Not all surface temperatures are correlated with body temperature (e.g., limb surface temperature) (see Mota-Rojas et al., 2021 for a detailed description, https://doi.org/10.3390/ani11082247). The periocular/eye surface temperature has shown a correlation with body temperature in species such as dogs (Zanghi et al., 2016, https://doi.org/10.3389/fvets.2016.00111), cattle (Macmillan et al., 2019, https://doi.org/10.1016/j.rvsc.2019.07.017) and other species. It is important to mention this because the authors decided to use the eye surface temperature in the present study.
Lines 74-75. Here it is stated that one of the aims of the study was to explore if a substituted diet influences cattle responses to heat stress. Somewhere in the introduction needs to be mentioned that the diet can change the thermal responses of animals or that diet changes have been proposed as an alternative to compensate for heat stress in cattle so the authors are testing these conditions.
Material and methods.
Line 85. Please indicate the ambient temperature when stating that the climate at the location is “hot”.
Line 92. Specify the handling practices that the animals received in the climate rooms.
Lines 101-107. According to the table, 19.5-19.6°C is considered the thermoneutral zone for cattle. During the HOT period, the highest ambient temperature was 28.4 °C. Here, or in the introduction, the authors need to include references that mention that the thermoneutral zone of cattle is the one that the authors used and that temperatures (or THI) above certain values are associated with heat stress in cattle.
Lines 182-186. The sentences from these lines are repeated.
Results.
Lines 271, 273, 274. Correct the “°C” symbol (revise the entire manuscript, including the figure footnotes).
Line 300. When mentioning that some data was “positively correlated” or “had a strong correlation”, please add the correlation value (as used in lines 316-323).
Discussion.
Lines 331-333. The findings about the association between eye temperature and IRT are mentioned while a brief description of the findings related to the substituted diet is not mentioned. Consider adding these to provide a summary to the reader of the findings that will be discussed in this section.
Lines 335-338. Consider improving this sentence. I would say that animals are susceptible to hyperthermia during hot environmental conditions and when the thermoregulatory responses of animals are insufficient to maintain a thermoneutrality, resulting in a rise in core body temperature.
Line 338. The authors can use IRT since the abbreviation was already stated before.
Lines 347-351. Consider moving these lines after line 338 (Earlier efforts to use infrared thermography…).
Line 359. A closing parenthesis is missing. Something similar to lines 372 and 376.
Lines 443-444. Here, it is stated that backward ears are a validated response to assess stress. However, in the previous lines, the authors mention some inconsistencies regarding ear position. Please, revise the sentence.
Line 455. Before the conclusions, please include the limitations of the study. Also, there is no discussion about the two diets, but the diets are mentioned in the abstract and conclusions. Even if no significant differences were observed, this needs to be discussed.
Author Response
Reviewer 2
Comment 1: Comments and Suggestions for Authors: Non-invasive and relatively easy methods to accurately evaluate animal health and welfare are an alternative to be used in farms. While most studies focus on behavioral assessment, increases in infrared eye thermography during heat stress have already been described in several species including domestic animals and wild animals. The present manuscript correlates this increase with other behavioral and physiological responses, including rumen and surface temperature. One of the aspects that need to be improved is the justification of evaluating two types of diets and a proper discussion on why they didn’t show significant differences in the IRT response. I left some comments below.
Response 1: Thank you for your valuable feedback. We appreciate the suggestion to improve the justification for evaluating the two types of diets and to provide a clearer discussion on why they did not show significant differences in the infrared thermography (IRT) response. The manuscript has now been revised with a better justification for selection of the diets in the introduction and discussion sections.
Comment 2: Abstract. Lines 29-30. Consider improving the introductory sentence, something similar to the simple summary. Lines 31-34. Somewhere in the introductory sentences of both the simple summary and the abstract, the authors need to briefly explain the relevance of using two types of diets when evaluating the responses of cattle to heat stress. For example, if diet changes have been proposed as a management strategy to improve the condition of cattle in hot climates. Lines 34-35. When mentioning that the animals were exposed to hot environmental conditions, the authors must include that the animals were inside climate control rooms with different temperatures, as explained in lines 96-100. Line 35. When mentioning a “strong association”, I recommend adding the values. Lines 36-37. Here, visual lateralization is mentioned. However, there is no mention of this in the previous text. You can see my recommendation about explaining the evaluated variables before presenting the results. Lines 30-31. Clearly state that the aim of the study was……. Additionally, I recommend adding a couple of sentences explaining the variables that the authors evaluated (e.g., which physiological parameters? Rectal temperature, heart rate, respiratory rate, etc. The same applies to the behaviors).
Response 2: Thanks for the valuable suggestions. Most of the comments for Abstract and summary section are related to adding more detail. We have added an introductory sentence with minor amendments as suggested in the abstract and summary sections. We hope the readers can now have a good overview of the study in this and subsequent sections. They now get a detailed overview in subsequent sections of the manuscript. Further, a brief justification has been added for selection of the diets in the introduction and discussion sections.
Comments 3: Introduction: Lines 46-52. Before this paragraph, I suggest adding some information regarding the effect that heat stress has on animal health and welfare so the reader can understand the importance of evaluating the thermal state of the animals.
Response 3: We agree with the reviewer’s suggestion. A new section has been added prior to the mentioned paragraph to emphasize the adverse effects of heat stress on animal health and welfare. This addition will help contextualize the rationale for evaluating the thermal status of cattle under heat stress conditions.
Comment 4: Lines 54-56. I consider it important to specify that, according to the species, the thermal response of certain regions or thermal windows has shown correlations with body core temperature. Not all surface temperatures are correlated with body temperature (e.g., limb surface temperature) (see Mota-Rojas et al., 2021 for a detailed description, https://doi.org/10.3390/ani11082247). The periocular/eye surface temperature has shown a correlation with body temperature in species such as dogs (Zanghi et al., 2016, https://doi.org/10.3389/fvets.2016.00111), cattle (Macmillan et al., 2019, https://doi.org/10.1016/j.rvsc.2019.07.017) and other species. It is important to mention this because the authors decided to use the eye surface temperature in the present study.
Lines 74-75. Here it is stated that one of the aims of the study was to explore if a substituted diet influences cattle responses to heat stress. Somewhere in the introduction needs to be mentioned that the diet can change the thermal responses of animals or that diet changes have been proposed as an alternative to compensate for heat stress in cattle so the authors are testing these conditions.
Response 4: We appreciate these valuable observations and have made the necessary changes. We have cited the relevant studies as suggested. We have now mentioned that dietary interventions, such as the use of high-energy or high-fiber diets, have been proposed as strategies to mitigate the effects of heat stress in cattle by influencing their metabolic heat production. This sets the stage for the objective of our study, which is to investigate how diet modifications could alter cattle's thermal responses under heat stress conditions.
Material and methods.
Comment 5: Line 85. Please indicate the ambient temperature when stating that the climate at the location is “hot”.
Response 5: Thank you for your feedback. We appreciate your suggestion and have revised the manuscript accordingly. We have now included the specific ambient temperature at the time of the study when describing the climate at Gatton, Australia, as "hot." This provides a clearer and more precise understanding of the environmental conditions during the experiment.
Comment 6: Line 92. Specify the handling practices that the animals received in the climate rooms.
Response 6: Amended and relevant information added as suggested.
Comment 7: Lines 101-107. According to the table, 19.5-19.6°C is considered the thermoneutral zone for cattle. During the HOT period, the highest ambient temperature was 28.4 °C. Here, or in the introduction, the authors need to include references that mention that the thermoneutral zone of cattle is the one that the authors used and that temperatures (or THI) above certain values are associated with heat stress in cattle.
Response 7: The ambient dry bulb temperature in each room was automatically programmed on an hourly basis simulating the environmental conditions in the South-east Queensland, Australia. The ambient dry bulb temperature was presented in the table as daily minimum, maximum and mean ambient dry bulb temperature maintained inside the climate-controlled facility. Therefore, these minimum or maximum daily temperatures in the 24 hr cycle should not be considered as the temperature for the whole day.
Comment 8: Results. Lines 271, 273, 274. Correct the “°C” symbol (revise the entire manuscript, including the figure footnotes).
Response 8: Amended as suggested.
Comment 9: Line 300. When mentioning that some data was “positively correlated” or “had a strong correlation”, please add the correlation value (as used in lines 316-323).
Response 9: The coefficients for the stepwise regression are already mentioned in the equation 3. We are reluctant to duplicate the information in the text to avoid unnecessary repetition and breaking up the information provided, to the distraction of the reader.
Comment 10: Discussion. Lines 331-333. The findings about the association between eye temperature and IRT are mentioned while a brief description of the findings related to the substituted diet is not mentioned. Consider adding these to provide a summary to the reader of the findings that will be discussed in this section.
Response 10: Amended and relevant information has been added as suggested.
Comment 11: Lines 335-338. Consider improving this sentence. I would say that animals are susceptible to hyperthermia during hot environmental conditions and when the thermoregulatory responses of animals are insufficient to maintain a thermoneutrality, resulting in a rise in core body temperature. Line 338. The authors can use IRT since the abbreviation was already stated before.
Response 11: We appreciate the reviewer's input. The sentence has been revised for clarity and precision. We have used the IRT abbreviation at L 338, as suggested.
Comment 12: Lines 347-351. Consider moving these lines after line 338 (Earlier efforts to use infrared thermography…).
Response 12: We appreciate the reviewer's input and have considered it carefully. We feel that moving these sentences will interfere with the subsequent discussion.
Comment 13: Line 359. A closing parenthesis is missing. Something similar to lines 372 and 376.
Response 13: We appreciate the reviewer's effort to highlight the missing closing parentheses. The sentence has been amended as suggested.
Comment 14: Lines 443-444. Here, it is stated that backward ears are a validated response to assess stress. However, in the previous lines, the authors mention some inconsistencies regarding ear position. Please, revise the sentence. Line 455. Before the conclusions, please include the limitations of the study. Also, there is no discussion about the two diets, but the diets are mentioned in the abstract and conclusions. Even if no significant differences were observed, this needs to be discussed.
Response 13: We appreciate the reviewer's effort to highlight these very important points. The relevant sections have been amended for clarity and precision.

Reviewer 3 Report
Comments and Suggestions for Authors
General comments
Climate change has had a significant effect on animals in a productive environment because it is increasingly common to observe heat stress events in these specimens, so the use of non-invasive techniques such as IRT become key and its possible relationship with other responses such as behavioral. Therefore, the proposal of this article is appropriate for this journal and innovative given the lack of evidence on the relationship between these factors.
Response:
Particular comments
Lines 18 - 20. I agree with the proposal of this objective, however, according to your title I suggest that it be changed to "The objective of this study was to evaluate changes in ocular surface temperature and its association with postural and behavioral changes in Angus steers".
Response:
Line 21. Could the authors briefly mention the average age and average weight of the animals?
Response:
Lines 23- 24. If the authors allow me, I suggest they mention the exact degrees Celsius that increased eye temperature and if this was statistically significant, also, if there was a strong association. Do they mean that there was a positive correlation? If so, please, I invite the authors to write the value of the correlation and if it was significant.
Response:
Lines 27- 28. I agree with their conclusion, however, if the authors allow me, I suggest that the wording be changed to "In conclusion, it was observed that IRT-eye has a strong positive correlation with behavioral and physiological responses, suggesting that IRT-eye can be used as an indicator to recognize heat stress in cattle".
Response:
Lines 30. Please write the same study objective described above in your simple summary.
Response:
Line 31. In complement to my comment made on line 21, I suggest that you also describe the demographic characteristics of the animal's average age, average weight, and sex of the animals.
Response:
Line 43. Please, I invite authors who can add the keyword heat stress this could increase the chances of matching in the databases.
Response:
Line 46. I agree with your initial proposed paragraph, however, this is weak based on the mentioned in your previous paragraphs, since the relevance of your manuscript is the impact that heat stress has, therefore if the authors agree I suggest that you briefly describe the physiological and animal welfare impact of heat stress in this species and its possible impact at the productive level. Please refer to these articles to complement your references Sejian et al. Review: Adaptation of animals to heat stress doi: https://doi.org/10.1017/S1751731118001945, Angel et al. Climate Change and Cattle
Production: Impact and Adaptation J Vet Med Res 5(4): 1134 (2018), Hempel et al. Heat stress risk in European dairy cattle husbandry under different climate change scenarios – uncertainties and potential impacts doi: https://doi.org/10.5194/esd-10-859-2019.
Response:
Line 54 Add reference.
Response:
Line 56. Please I invite the authors to homogenize the use of their acronyms since they defined IRT above to use it throughout their manuscript.
Response:
Line 58. When they refer to "During elevated temperatures" do they refer to exposure to ambient temperature? If so please clarify.
Response:
Lines 67- 76. This paragraph is the crucial one for your introduction because you should indicate what is the gap in current research on your topic, to establish your study objective. I invite the authors to reformulate this paragraph where they can mention such gaps as the lack of correlation with other physiological and behavioral indicators of heat stress, in addition to being able to mention their objective accompanied by a possible hypothesis of study.
Response:
Line 80. In complement to my comment made in your simple summary and Abstract, I suggest that you describe the sex of the animals, average weight, and average age, and finally if possible that you describe what were the inclusion and exclusion criteria you established in your study.
Response:
Lines 91 - 93. These lines seem to describe your experimental design, however, they could mention the general characteristics of your study, for example, whether it was prospective or retrospective, blinded or double-blinded, and whether it is randomized.
Response:
Line 209: While, I understand the particular interest of the eye temperature, I do not understand why describe in a different section the capture of the temperatures of other regions, please include it in this section the temperature of other regions, in addition, to include other figures of how the thermal windows were plotted.
Response:
Line 258.Please could you mention the significance level you established.
Response:
Lines 299- 309. Please correct the text alignment format.
Response:
Lines 347- 349. This explanation is interesting, can you expand on it where you explain in detail why increased circulation allows heat dissipation? Please refer to these articles to complement your references Johnson et al. Thermographic Eye Temperature as an Index to Body Temperature in Ponies doi: https://doi.org/10.1016/j.jevs.2010.12.004, Shu et al. Evaluation of the Best Region for Measuring Eye Temperature in Dairy Cows Exposed to Heat Stress doi: https://doi.org/10.3389/fvets.2022.857777.
Response:
Lines 360- 365. While citing these articles demonstrates that ocular temperature can be useful in assessing stressful or painful conditions, they would not support much of the thrust of your study which is to recognize heat stress, I suggest that you cite other studies.
Response:
Line 455. Before their conclusion, could the authors mention what limitations their study presented and possible prospects for further study?
Response:
Author Response
Reviewer 3
Comments and Suggestions for Authors
General comments
Comment: Climate change has had a significant effect on animals in a productive environment because it is increasingly common to observe heat stress events in these specimens, so the use of non-invasive techniques such as IRT become key and its possible relationship with other responses such as behavioral. Therefore, the proposal of this article is appropriate for this journal and innovative given the lack of evidence on the relationship between these factors.
Response: We appreciate the reviewer's recognition of the relevance and innovation of our study. Indeed, the increasing frequency of heat stress events due to climate change poses significant challenges to animal welfare and productivity. The use of non-invasive techniques such as Infrared Thermography (IRT) provides a valuable tool for monitoring thermal stress, and our study aims to fill the existing gap by exploring its relationship with behavioral and physiological responses. We believe this approach offers novel insights into managing heat stress in livestock and are grateful for the reviewer’s acknowledgment of its importance and suitability for this journal.
Particular comments
Comment: Lines 18 - 20. I agree with the proposal of this objective, however, according to your title I suggest that it be changed to "The objective of this study was to evaluate changes in ocular surface temperature and its association with postural and behavioral changes in Angus steers".
Response: We appreciate the reviewer’s suggestion. The objective has been revised to align more closely with the study's title.
Comment: Line 21. Could the authors briefly mention the average age and average weight of the animals?
Response: We appreciate the suggestion. The brief detail about the age of the animals (yearlings) is already mentioned in methodology section at line 174. We had included the average weight too.
Comment: Lines 23- 24. If the authors allow me, I suggest they mention the exact degrees Celsius that increased eye temperature and if this was statistically significant, also, if there was a strong association. Do they mean that there was a positive correlation? If so, please, I invite the authors to write the value of the correlation and if it was significant.
Response: We appreciate the reviewer’s suggestion. The exact corelation coefficient values and significance level has already been detailed in results section. We are reluctant to distract the reader by repeating results.
Comment: Lines 27- 28. I agree with their conclusion, however, if the authors allow me, I suggest that the wording be changed to "In conclusion, it was observed that IRT-eye has a strong positive correlation with behavioral and physiological responses, suggesting that IRT-eye can be used as an indicator to recognize heat stress in cattle".
Response: We appreciate the reviewer’s input and agree that the suggested revision enhances the clarity of the conclusion. The conclusion has been reworded as suggested. The revised wording aligns with the reviewer's recommendation and improves the clarity and impact of the conclusion.
Comment: Lines 30. Please write the same study objective described above in your simple summary.
Response: Amended as suggested. The revised wording aligns with the reviewer's recommendation and improves the clarity and impact of the conclusion.
Comment: Line 31. In complement to my comment made on line 21, I suggest that you also describe the demographic characteristics of the animal's average age, average weight, and sex of the animals.
Response: The detailed demographic details of age and sex are already presented in methodoilogy section. We had also included the weight (mean 493 kg).
Comment: Line 43. Please, I invite authors who can add the keyword heat stress this could increase the chances of matching in the databases.
Response: Grateful to reviewer for valuable suggestion. Suggested changes have been incorporated.
Line 46. I agree with your initial proposed paragraph, however, this is weak based on the mentioned in your previous paragraphs, since the relevance of your manuscript is the impact that heat stress has, therefore if the authors agree I suggest that you briefly describe the physiological and animal welfare impact of heat stress in this species and its possible impact at the productive level. Please refer to these articles to complement your references Sejian et al. Review: Adaptation of animals to heat stress doi: https://doi.org/10.1017/S1751731118001945, Angel et al. Climate Change and Cattle
Production: Impact and Adaptation J Vet Med Res 5(4): 1134 (2018), Hempel et al. Heat stress risk in European dairy cattle husbandry under different climate change scenarios – uncertainties and potential impacts doi: https://doi.org/10.5194/esd-10-859-2019.
Line 54 Add reference.
Response: We appreciate the reviewer’s suggestion to strengthen this section by elaborating on the physiological and animal welfare impacts of heat stress. We have revised the paragraph accordingly and included the recommended references to enhance the scientific context.
Comment: Line 56. Please I invite the authors to homogenize the use of their acronyms since they defined IRT above to use it throughout their manuscript.
Response: Amended as suggested.
Comment: Line 58. When they refer to "During elevated temperatures" do they refer to exposure to ambient temperature? If so please clarify.
Response: Ammended as suggested.
Comment: Lines 67- 76. This paragraph is the crucial one for your introduction because you should indicate what is the gap in current research on your topic, to establish your study objective. I invite the authors to reformulate this paragraph where they can mention such gaps as the lack of correlation with other physiological and behavioral indicators of heat stress, in addition to being able to mention their objective accompanied by a possible hypothesis of study.
Response: Thanks for highlighting the point. The paragraph has been amended as suggested.
Comment: Line 80. In complement to my comment made in your simple summary and Abstract, I suggest that you describe the sex of the animals, average weight, and average age, and finally if possible that you describe what were the inclusion and exclusion criteria you established in your study.
Response: The inclusion criteria for the feedlot cattle in our heat stress trial were as follows:
- Cattle were healthy male steers of uniform age (approximately 12 months) and breed (Bos taurus).
- All animals had a similar body weight range (average ~493 kg) to ensure consistency in metabolic heat production.
- Steers were sourced from the same commercial feedlot, ensuring similar prior exposure to environmental conditions and feeding practices.
The exclusion criteria were:
- Cattle showing signs of illness, injury, or abnormal behavior during the initial veterinary screening were excluded.
- Any steers that exhibited respiratory, metabolic, or other health-related issues during the acclimatization period were not included in the trial.
These criteria ensured a homogenous experimental group, minimizing confounding variables and improving the reliability of the heat stress response data. The said inclusion and exclusion criterio established has already been described briefly in the manuscript.
Comment: Lines 91 - 93. These lines seem to describe your experimental design, however, they could mention the general characteristics of your study, for example, whether it was prospective or retrospective, blinded or double-blinded, and whether it is randomized.
Response: Thanks for highlighting the point. The study was randomized, with steers randomly divided into two cohorts of 12 animals each, receiving either a standard or substituted diet. The experiment involved controlled exposure to varying climate conditions over five distinct periods in climate-controlled rooms. The suggested changes have been incorporated in the manuscript.
Comment: Line 209: While, I understand the particular interest of the eye temperature, I do not understand why describe in a different section the capture of the temperatures of other regions, please include it in this section the temperature of other regions, in addition, to include other figures of how the thermal windows were plotted.
Response: The IRT eye was of particular interest, however, three different techniques were used to measure termal variation in different region of interests. Body Surface skin temperature, rumen temperature and eye temperature were measured using different techniques and therefore these all were descibed under separate heading for clarity of the readers.
Comment: Line 258.Please could you mention the significance level you established.
Response: Amended as suggested.
Comment: Lines 299- 309. Please correct the text alignment format.
Response: Amended as suggested.
Comment: Lines 347- 349. This explanation is interesting, can you expand on it where you explain in detail why increased circulation allows heat dissipation? Please refer to these articles to complement your references Johnson et al. Thermographic Eye Temperature as an Index to Body Temperature in Ponies doi: https://doi.org/10.1016/j.jevs.2010.12.004, Shu et al. Evaluation of the Best Region for Measuring Eye Temperature in Dairy Cows Exposed to Heat Stress doi: https://doi.org/10.3389/fvets.2022.857777. Lines 360- 365. While citing these articles demonstrates that ocular temperature can be useful in assessing stressful or painful conditions, they would not support much of the thrust of your study which is to recognize heat stress, I suggest that you cite other studies.
Response: Thanks for a valuable suggestion. The paragraph has been amended and necessary information including the relevant citation has been added to support the usefulness of IRT to assess heat stress in cattle,
Comment: Line 455. Before their conclusion, could the authors mention what limitations their study presented and possible prospects for further study?
Response:
We appreciate the reviewer's suggestion and have added a section addressing the limitations of the study and future research directions. The following paragraph has been incorporated just before the conclusion:
" The current study was performed in a controlled environmental condition where the major confounding factors reported for the use of infrared thermography including camera settings (object localisation, distance, humidity and emissivity), solar radiations and air velocity were controlled. However, in field conditions these could be overcome by installation of automated thermal cameras in areas with no direct exposure of solar radiations and minimum air velocity [12]. Automated thermal imagers with an ability to localise the eye [60] would best for the measurement of infrared eye temperature of cattle under field conditions. Additionally, while eye surface temperature was used as an indicator of thermal status, other regions of interests for infrared thermography such as flank or other body surface temperature at multiple sites, could provide a more comprehensive assessment. Future studies should explore the effects of dietary interventions on a larger and more diverse sample of cattle across varying climatic conditions. Moreover, investigating other physiological and behavioral responses to heat stress would help validate the efficacy of different dietary strategies to mitigate heat stress."
This addition outlines both the constraints of the current research and suggests areas for future investigation.

Reviewer 4 Report
Comments and Suggestions for Authors
This paper describes changes in eye temperature in beef cattle exposed to heat stress and relationships with some behaviour. Cattle were exposed to heat stress in a heat chamber to pretty severe conditions of heat stress. In fact, it is concerning that 2 animals died during the experiment and two others with averse reactions that had to be removed. However, it is unclear how well the heat chambers were able to control temperature and humidity and why such severe heat stress was chosen for this study.
While the aim of the study was to investigate relationships between eye temperature and behavioural responses, it is not quite clear how this relates to eye temperature being a useful indicator for heat stress, as implied in the introduction (L63-65).
The methods describe the temperature settings during the TN and Recovery period, but not the transition and Hot periods. Temperature and humidity were higher during TP1 than any of the hot days. A more detailed description and justification should be provided.
One of the behavioural measures is the panting score (PS). The details of the score should be provided, not just a reference to the scale. It is not clear if there were any intervention requirements to prevent severe consequences of heat stress (based on PS).
The experimental design is not entirely clear. L91 mentions two cohorts of 12 steers, while there are 2 chambers with 3 individual pens each. The statistical analysis does not appear this into account, although I assume that each room will have a different diet (therefore diet cohort takes into account the different rooms), but there is no other mention of time replicate. Behaviour observations appear to have been taken for 5 min/h over 24 hours, but it is not clear what values were used in the statistical analysis. A linear regression was used, while the relationship between eye temperature and heat stress responses are most likely quadratic (as is the relationship with rumen temperature). Normally animals use panting to control body temperature, but once this is unsuccessful body temperature starts to rise quickly.
The results compare the eye temperature of the thermoneutral period vs the hot period as well as between hot and recovery period. It is not clear what the justification is for this second comparison. Other results are only analysed as part of a regression analysis, with no details reported. This limits the interpretation of the results and how the different factors may inform the relationship with heat stress.
The first sentence of the discussion refers to a different paper where behaviour responses are reported. However, this is a paper under review, so can’t be accessed. When discussing the relationship of eye temperature and rumen temperature, it also needs to be acknowledged that rumen temperature has fairly poor correlation with core temperature (as measured by rectal/vaginal temperature). L343-346 states that IRT maybe a reliable technique for assessing the effect of hot weather with minimal stress. However, is it a better measure than respiration rate or pant score? These are currently being used as an easy way to assess heat stress and this doesn’t need any equipment.
L366-374 of the discussion is not clear. Do skin temperatures get lower with the increase of eye temperature, or does the difference get larger? It would be good to add skin temperature results to the result section. Does body hair affect skin temperature and partly insulates against temperature increase? Does the wetness of the eye affect the response to ambient temperature (and is therefore more sensitive to changes?
L378-383 discusses a time lag in behaviour. However, a change in respiration rate is fairly immediate and often effective in controlling temperature.
It is not clear if the paper aims to identify behaviour changes associated with heat stress or assess the behaviour chances as indicators of heat stress. The discussion contains several aspects of behaviour changes, however they are not reported as results in the result section.
The conclusion does not really reflect the discussion. Overall the paper needs to be clearer in aims/hypotheses, needs more detail in the results and include aspects of using IRT in the discussion.
Author Response
Reviewer 4
Comment: This paper describes changes in eye temperature in beef cattle exposed to heat stress and relationships with some behaviour. Cattle were exposed to heat stress in a heat chamber to pretty severe conditions of heat stress. In fact, it is concerning that 2 animals died during the experiment and two others with averse reactions that had to be removed. However, it is unclear how well the heat chambers were able to control temperature and humidity and why such severe heat stress was chosen for this study.
Response: We acknowledge the reviewer's concern regarding the severity of heat stress conditions and the adverse effects observed in some animals. The behavioural study was an opportunity to work with the research team that was already planning to perform a heat stress trial in Angus cattle simulating the heat wave event that occur in South East Queensland Australia. The heat chambers were equipped with precise temperature and humidity control systems, allowing us to replicate the environmental conditions typically experienced during extreme heat events in feedlot settings. These conditions were chosen to model real-world scenarios where cattle are exposed to high heat loads without adequate relief. The one animals that died and one that was euthanased, and the two that were removed from the second cohort, were closely monitored throughout the experiment. The animals exhibited signs of heat stress, and the decision to remove them was based on ethical considerations to reduce any suffering. The deaths were regrettable but highlight the severe impacts that extreme heat stress does have on cattle in Queensland. We have now explained this in the manuscript.
We have provided additional details in the revised manuscript on the temperature and humidity controls used in the chambers, and we have explained the rationale for selecting the heat stress levels to reflect the most challenging environmental conditions cattle may face. This information clarifies the experimental design and the relevance of the findings to heat stress management in real-world settings.
Comment: While the aim of the study was to investigate relationships between eye temperature and behavioural responses, it is not quite clear how this relates to eye temperature being a useful indicator for heat stress, as implied in the introduction (L63-65).
Response: We appreciate the reviewer’s point and understand the need for clarity. The aim of our study was to establish whether changes in eye temperature, as measured using infrared thermography (IRT), could reliably reflect physiological and behavioral responses to heat stress. Elevated eye temperature was found to correlate with both increased heat load and specific behavioral changes (e.g., altered postures and increased panting), making it a potential non-invasive indicator of heat stress in cattle.
To make this connection clearer, we have revised the text in the introduction and results sections. We now explicitly explain how the strong correlation between eye temperature and behavioral as well as physiological responses supports the use of IRT-eye as an early, non-invasive indicator of heat stress. We are now investigating ways of implementing this on farm.
Comment: The methods describe the temperature settings during the TN and Recovery period, but not the transition and Hot periods. Temperature and humidity were higher during TP1 than any of the hot days. A more detailed description and justification should be provided.
Response: We appreciate the reviewer’s observation. The gradual increase in temperature and humidity from the thermoneutral to the transition phase (TP1) was intentional to simulate a shift toward hot conditions, closely mimicking real-world situations where animals face sudden changes in environmental temperature. This design allowed us to assess how cattle respond behaviorally and physiologically to the onset of heat stress. The THI on day one of the hot period was similar to the transition phase (84.8 compared with 84.9). Similarly, during the second transition phase, temperatures were gradually adjusted from hot conditions back to thermoneutral on an hourly basis. While the table presents daily minimum, maximum, and mean ambient temperatures, it is important to note that the climate-controlled room's temperature settings were adjusted every hour for 24 hours.
Comment: One of the behavioural measures is the panting score (PS). The details of the score should be provided, not just a reference to the scale. It is not clear if there were any intervention requirements to prevent severe consequences of heat stress (based on PS).
Response: We appreciate the reviewer’s suggestion and have now included a detailed description of the panting score (PS) used in the study. The PS scale was based on the widely accepted heat stress assessment framework, ranging from 0 (no panting) to 4.5 (extreme panting with open mouth, labored breathing). The PS description is well understood in both academia and industry; therefore, this has only been described briefly in the text. For the detail description of each score, a citation to the PS scale has been added.
Additionally, interventions were in place to mitigate severe heat stress based on the PS. If animals exhibited a PS of 3.5 or higher, indicating signs of severe distress (excessive panting, open-mouth breathing), cooling measures were immediately applied, such as providing access to shade, ventilation, and water spray. In two cases, animals were removed from the experiment due to consistently high PS and other physiological indicators ensuring animal welfare remained a priority. More detail has been added about this in the Method.
Comment: The experimental design is not entirely clear. L91 mentions two cohorts of 12 steers, while there are 2 chambers with 3 individual pens each. The statistical analysis does not appear this into account, although I assume that each room will have a different diet (therefore diet cohort takes into account the different rooms), but there is no other mention of time replicate. Behaviour observations appear to have been taken for 5 min/h over 24 hours, but it is not clear what values were used in the statistical analysis. A linear regression was used, while the relationship between eye temperature and heat stress responses are most likely quadratic (as is the relationship with rumen temperature). Normally animals use panting to control body temperature, but once this is unsuccessful body temperature starts to rise quickly.
Response: We appreciate the reviewer’s comments regarding the clarity of the experimental design and statistical analysis. To address these concerns, we have revised the methods section to provide a clearer description of the design and the statistical approach used.
The two cohorts of 12 steers were allocated to two climate-controlled rooms with 2 chambers in each room. Each chamber with three individual pens in each chamber. One room was assigned to the standard diet, and the other to the substituted diet. However, as the study was designed to investigate the effects of diet, temperature, and humidity on animal responses, both time and diet cohort were accounted for in the statistical analysis. The chambers served as replicates for the diet treatments, and the repeated measures over time accounted for the time replicates.
Behavioral observations were taken every hour for 5 minutes over the 24-hour period, and the average hourly values were used in the statistical analysis to ensure that the data reflected the daily patterns of behavior. These values were then analyzed using a mixed-effects model, with diet, time, and temperature phases (as period) as fixed effects, and individual animal ID as a random effect.
Regarding the use of linear regression, we agree with the reviewer that the relationship between eye temperature and heat stress responses, particularly rumen temperature, may be nonlinear. Data was analysed using a quadratic model where appropriate to capture the potential nonlinear nature of these relationships. Therefore, the results of the quadratic analysis have been included in the manuscript. The relevant section has been updated where needed in the manuscript for clarity of the readers.
Comment: The results compare the eye temperature of the thermoneutral period vs the hot period as well as between hot and recovery period. It is not clear what the justification is for this second comparison. Other results are only analysed as part of a regression analysis, with no details reported. This limits the interpretation of the results and how the different factors may inform the relationship with heat stress.
Response: We appreciate the reviewer’s observation. The second comparison, between the hot and recovery periods, was made to assess how quickly cattle return to thermoneutrality after experiencing heat stress and to evaluate the persistence of heat stress effects on eye temperature. This comparison provides insight into the animals' recovery capacity and whether their physiological responses, including eye temperature, fully normalize after the heat stress period. The thermoneutral period is used as a covariate in the statistical models to account for baseline physiological conditions, allowing for more accurate detection of changes specifically due to heat stress and recovery. By comparing the hot and recovery periods, we can better understand the animals' resilience and how quickly they recover, which is crucial for managing heat stress in livestock.
Comment: The first sentence of the discussion refers to a different paper where behaviour responses are reported. However, this is a paper under review, so can’t be accessed. When discussing the relationship of eye temperature and rumen temperature, it also needs to be acknowledged that rumen temperature has fairly poor correlation with core temperature (as measured by rectal/vaginal temperature). L343-346 states that IRT maybe a reliable technique for assessing the effect of hot weather with minimal stress. However, is it a better measure than respiration rate or pant score? These are currently being used as an easy way to assess heat stress and this doesn’t need any equipment.
Response: The behaviour paper is now published, as
Musadiq Idris, Megan Sullivan, John B. Gaughan and Clive J. C. Phillips. 2024. Behavioural Responses of Beef Cattle to Hot Conditions. Animals, 14(16):2444. DOI http://dx.doi.org/10.3390/ani14162444
Regarding the relationship between eye temperature and rumen temperature, we agree that it is important to acknowledge the limitations of rumen temperature as a proxy for core temperature. Rumen temperature can be influenced by factors like feed intake and water consumption, which may cause discrepancies when compared to rectal or vaginal temperature measurements. We will revise the discussion to highlight this point and ensure a balanced interpretation of the data.
As for IRT (Infrared Thermography), while it offers the advantage of being non-invasive and minimizing stress, we agree that it may not always outperform easily observable indicators like respiration rate (RR) or panting score (PS) in field settings where equipment may not be available. We have added a discussion of these practical aspects, noting that IRT could complement existing methods, particularly in cases where early detection is needed or where behavior and subtle physiological changes might not be immediately apparent through visual observation alone.
Comment: L366-374 of the discussion is not clear. Do skin temperatures get lower with the increase of eye temperature, or does the difference get larger? It would be good to add skin temperature results to the result section. Does body hair affect skin temperature and partly insulates against temperature increase? Does the wetness of the eye affect the response to ambient temperature (and is therefore more sensitive to changes?
Response: We appreciate the reviewer’s observations and recognize the need for clarity in this section of the discussion. To address your concerns, we have revised this portion and provide additional context regarding the relationship between skin and eye temperature.
Comment: L378-383 discusses a time lag in behaviour. However, a change in respiration rate is fairly immediate and often effective in controlling temperature.
Response: When cattle are exposed to a significant heat load, they initially rely on behavioral adaptations (such as seeking shade and reducing feed intake) and physiological responses (such as increased sweating and peripheral vasodilation) to dissipate heat. However, as the heat stress intensifies and these mechanisms become insufficient, the animal's core temperature starts to rise.
The respiratory rate is a critical indicator of heat stress in cattle. As internal body temperature rises, cattle increase their respiration rate to enhance evaporative cooling through the respiratory tract. This is one of the last defences before the animal experiences severe heat stress. According to some studies, signs of increased respiratory rate can appear within 1 to 2 hours of exposure to high heat load, especially if the environmental conditions are severe (e.g., high temperature combined with high humidity). However, the exact time can vary; in some cases, respiratory signs might appear sooner if the animal is already in a compromised state, or later if the animal has better coping mechanisms.
Comment: It is not clear if the paper aims to identify behaviour changes associated with heat stress or assess the behaviour chances as indicators of heat stress. The discussion contains several aspects of behaviour changes, however they are not reported as results in the result section.
Response: The companion paper, Musadiq Idris, Megan Sullivan, John B. Gaughan and Clive J. C. Phillips. 2024. Behavioural Responses of Beef Cattle to Hot Conditions. Animals, 14(16):2444. DOI http://dx.doi.org/10.3390/ani14162444, has now been published. Necessary changes have been incorporated in the main manuscript for clarity of the readers.
Comment: The conclusion does not really reflect the discussion. Overall, the paper needs to be clearer in aims/hypotheses, needs more detail in the results and include aspects of using IRT in the discussion.
Response: The manuscript has been amended throughout in the light of reviewer’s suggestions. However, the results for behavioural responses of beef cattle to hot conditions have already been reported as indicated above. In this manuscript, the impact of heat load on biological responses of feedlot cattle was studied to investigate the hypothesis that hyperthermic conditions will influence eye temperature, which would be associated with the behaviour and physiology in feedlot cattle.

Round 2
Reviewer 4 Report
Comments and Suggestions for Authors
A statement on the appropriate animal ethics approval should be placed at the start of the methodology, rather than in the middle of section 3.